# Determinants of neonatal mortality among hospitalised neonates with sepsis at Queen Elizabeth Central Hospital, Blantyre, Malawi: A mixed-methods study

Lucky Mangwiro[1,2]*, Joseph Misyenje[1], Pui-Ying Iroh Tam[3,4,5‡], Kondwani Kawaza[1,4‡], Alinane Linda Nyondo Mipando[2,3,6‡]

1 NEST 360 Program, Kamuzu University of Health Sciences, Blantyre, Malawi, 2 Department of Health Systems and Policy, Kamuzu University of Health Sciences, Blantyre, Malawi, 3 Paediatrics and Child Health Research Group, Malawi Liverpool and Wellcome Trust, Blantyre, Malawi, 4 Department of Paediatrics and Child Health, Kamuzu University of Health Sciences, Blantyre, Malawi, 5 Department of Clinical Sciences, Liverpool School of Tropical Medicine, Liverpool, England, United Kingdom, 6 Department of Women's and Children's Health, University of Liverpool, Liverpool, England, United Kingdom

‡ PI, KK, and ALNM are joint last authors.
* luckyngwiro@gmail.com

## Abstract

Neonatal sepsis-related deaths remain a significant health problem contributing to higher morbidity and mortality rates, particularly in low resource settings, such as Malawi. However, there is limited information to associate risk factors and health system factors with mortality. This study investigated the risk factors associated with mortality and explored health system factors contributing to deaths among neonates with sepsis at Queen Elizabeth Central Hospital (QECH), Blantyre, Malawi. This mixed-method study utilised a convergent parallel approach to describe the determinants of neonatal mortality among neonates with sepsis. We selected this design because it allowed the researchers to triangulate, support and enhance the internal and external validity of the results. We retrospectively reviewed 237 neonatal records using a simple random sampling technique for cross-sectional quantitative data. Exploratory qualitative data was collected using a semi-structured interview guide from 10 purposively selected healthcare workers directly involved in providing neonatal care through in-depth interviews. Quantitative data were analysed using univariate and multivariate logistic regression in Stata v.14; qualitative data were analysed manually using a thematic analysis approach. We found that gestation age (OR 0.76 (95% CI: 0.58, 0.99), p-value = 0.040) and number of days spent in the hospital (OR 0.64 (95% CI: 0.48, 0.85), p-value = 0.002) were the most predictive risk factors. The qualitative inquiry showed the maternal behavioural factors; reporting late to hospital, cultural and religious beliefs; maternal health related factors: prolonged labour, unnecessary vaginal examinations, premature rupturing of membranes; Neonatal factors: prematurity, meconium aspiration, home deliveries and lastly, health system factors included delay in treatment, referrals and blood culture results, limited resources contributed to documented clinical outcomes. Determinants of neonatal mortality were gestation age, number of days spent in the hospital,

**Data Availability Statement:** The datasets used and analysed during the current study will be available without any restrictions.

**Funding:** This research is funded by the National Institute for Health Care Research (NIHR) grant number NIHR131237 using United Kingdom (UK) aid government to support global health research through Kamuzu university of health sciences Innovative Management Practices to Enhance hospital quality and save lives in Malawi (IMPRESS) project. The views expressed in this publication are those of the authors and not necessarily those of the NHIR or the UK government.

**Competing interests:** The authors have declared that no competing interests exist.

**Abbreviations:** ANC, Antenatal Care; CI, : Confidence Interval; NCU, Neonatal Care Unit; OR, Odds Ratio; PROM, Premature Rupture of Membranes; QECH, Queen Elizabeth Central Hospital; HCW, Healthcare workers.

maternal behavioural and health related, neonatal and health system factors. Reducing mortality among neonates with sepsis will require a multi-sectoral approach.

## Introduction

Neonatal sepsis is a significant global health concern due to the high mortality rate and substantial economic costs associated with it [1]. Neonatal sepsis is characterised by inflammation in newborns, regardless of whether they have an infection [2]. Globally, 1.3 million annual incident cases of neonatal sepsis are responsible for causing 203,000 neonatal deaths [3]. In sub-Saharan Africa, sepsis-related neonatal mortality rates are high and range between 17.0 to 29.0% [4]. Malawi has a relatively high neonatal sepsis incidence of 23.6%, the second cause of mortality among neonates [5].

Evidence suggests a need for more geographically specific data illustrating sepsis risk factors. Previous studies have focused on cause-specific neonatal mortality [6], sepsis and its aetiology [7] assessing the quality of newborn care [8], newborn survival in Malawi [9], hypothermia [10], birth asphyxia [11], stillbirths, early neonatal deaths (NND) [12], and caregivers' perspectives on hand hygiene [13]. Other studies that focused on health system factors found that the staff struggled to implement infection prevention (IP) procedures to their fullest due to a large number of patients, poor hand hygiene practices, lack of soap, disinfectants and linen, and an unreliable water supply for caregivers and healthcare workers (HCW) made problems worse [8]. It has been shown that unclean cord care and lack of hand washing before handling newborns could put them at risk for sepsis [13].

Research conducted in a rural district of eastern Uganda found that HCW also encountered difficulties in preventing the spread of infection to newborns within the facility due to understaffed and lack of essential or basic lifesaving equipment such as resuscitation kits for newborns [14]. Furthermore, delivering at a health facility is sometimes not aided by qualified and skilled personnel. Therefore, the care given is inadequate. Studies done in low-income countries like Pakistan have also indicated that sub-standard care, inadequate training, low staff competence and a lack of resources, including equipment and medication, contribute to neonatal deaths [14]. Information about health system factors such as availability of training, delayed referrals, blood culture results and treatment, lack of monitoring, and negligence linked to sepsis and neonatal mortality in Malawi is limited.

The Newborn Essential Solutions and Technology (NEST) 360 package aims to prevent and treat the three major causes of newborn death in Africa: preterm birth, asphyxia, and infection. As part of NEST 360's goal to reduce preventable neonatal deaths, 38 facilities, including Queen Elizabeth Central Hospital (QECH) neonatal care unit (NCU), received Pumani bCPAP, photo therapy lights, radiant warmers, syringe pumps, oxygen concentrators, suction machines, birilubinometers, hemoglobinometers, pulse oximeters, glucometers, and baby coats. Out of 4,308 neonatal admissions from May 2018 to June 2019, 1,244 (28.9%) had suspected early-onset sepsis (EOS) [15]. NEST 360 provided data for all newborns admitted to Malawi's Neonatal Intensive Care Unit. Out of all the newborns admitted to the NCU at QECH in Malawi in 2021, 12.6% of the newborns were diagnosed with neonatal sepsis. Given the high neonatal sepsis-related mortality, it is imperative to understand the contributory risk and health system factors. Therefore, this study aimed to assess factors associated with mortality risk and explore HCW perspectives on factors contributing to neonatal deaths among hospitalised newborns with sepsis in Malawi.

## Methods

### Study setting

This study was conducted in the Chatinkha NCU at QECH, Blantyre, Malawi. QECH is a tertiary facility [16] situated in the southern region with a population of approximately 5 million; it is a referral hospital and offers specialised healthcare with paediatric specialists, registrars, intern medical doctors, and midwives. On each shift, there are two to four nurse midwives [13]. The NCU has 60 beds, divided into high-risk and low-risk areas according to need, and facility-based care of small and sick neonates, from neonatal resuscitation to kangaroo mother care (KMC), bubble continuous positive airway pressure (bCPAP), intravenous (IV) fluids, feeding support, oxygen, sepsis management, antibiotics, and phototherapy [17]. In 2021, QECH NCU had 3,713 neonatal sepsis admissions. Out of 3,713, there were 470 neonatal sepsis deaths. In 2021, the QECH NCU admitted a total of 3,717 newborns, with monthly admissions ranging from 170 (March) to 377 (November). The mortality rates varied significantly throughout the year, from a low of 1.2% in March to a high of 20.9% in December, with a total of 470 deaths recorded for the year.

### Study design

A convergent parallel mixed-method study was conducted, and it employed cross-sectional quantitative and exploratory qualitative approaches [18]. The quantitative component involved a retrospective review of 237 records of neonates aged 0–28 days admitted with a diagnosis of clinically suspected neonatal sepsis and neonates with confirmed blood culture positive receiving care at NCU between 1 January to 31 December 2021. Qualitative data were collected from 25 January 2023 to 13 February 2023 using a semi-structured interview guide that was developed. They utilised an exploratory approach comprising eight in-depth interviews and two key informant interviews with HCWs who provide care to neonates. A convergent parallel design was selected because it allowed the researchers to triangulate, validate and support the results relative to the same phenomenon and enhance internal and external validity [19]. While quantitative approaches measured the strength of association between variables of interest, the qualitative data explored HCW perceptions on health system factors contributing to mortality among neonates with sepsis, complementing the quantitative findings.

### Recruitment, sampling and data collection

**Quantitative sampling.**   We retrospectively reviewed the records of recruited neonates aged 0–28 days admitted to the NCU based on the final suspected or confirmed neonatal sepsis diagnosis. To accurately represent the entire population and give all participants an equal chance of being chosen, neonatal records from the NCU at QECH, the sample size was determined using Cochran's formula: $n = z^2pq/e^2$, where n is the minimum sample size needed, p is the estimated prevalence rate of neonatal mortality due to sepsis in Malawi (0.19), e is the error rate (0.05), q is 1-p, and z is the standard variable corresponding to the 95% confidence interval (1.96). The calculation yielded a sample size of 236.48, which was rounded up to 237 [20]. Power calculation was not done. To select these records, we used a simple random sampling technique. Numbers were assigned to eligible participants in the sampling frame, and the RANDBETWEEN (1, N) function in Microsoft Excel was used to randomly select the participants' records for inclusion in the study. Data were collected by the researcher using a structured questionnaire which included the following variables; mothers age, residence, maternal chronic conditions, antenatal corticosteroids given, VDRL status, HIV status, type of birth, sex, birth location, reason for admission, birth weight, gestational age, child age, phototherapy,

mode of delivery, blood culture results, vital signs checked (temperature, respiratory rate, heart rate, oxygen saturation), antibiotics given, cord chlorhexidine administered, blood glucose test done, number of days spent in the hospital and discharge outcome. Data collected were entered into Microsoft Excel.

**Qualitative sampling.**    A purposive sampling technique was used to select study participants with characteristics of interest in the qualitative component [19]. Ten NCU HCWs were willing to share their experiences and were purposively selected for the study. The non-probability sampling method was deliberately chosen to have a sample that contains characters of interest [19]. The perspectives of HCWs on factors contributing to neonatal deaths were clarified through two key informants and eight in-depth interviews. HCWs willing to participate in the study were scheduled according to their free time. The researcher conducted interviews after explaining the study's purpose to each participant. Those willing to participate in the study provided written informed consent, and interviews were conducted using a semi structured interview guide ("S1 Appendix") in the NCU Matrons' office. The questions were developed and informed with an analytical framework for the study of child survival in developing countries by Mosley et al that was modified to the conceptual framework of factors associated with neonatal mortality in Malawian setting based on the available information in the sampled facility. The variables for the risk factors were categorized into three: distal factors included socio-economic and demographic variables; intermediate factors included maternal reproductive and health delivery factors; and proximal factors included neonatal factors. This included questions about general knowledge of risk factors and management of neonatal sepsis, formal training on neonatal sepsis, IP measures followed in the unit and challenges faced when managing neonatal sepsis cases. Data were recorded in English and Chichewa using a password-protected digital recorder. The interviews, on average, lasted 17 minutes, ranging from 13 to 26 minutes. A password-protected computer and external hard drive were used to transfer, store, and back up digital voice recorder data. The supervisors cross-checked and reviewed the extracted audio and transcripts for completeness, verified that the information captured was accurate, clarified any errors, and corrected any errors to minimise threats to the study's credibility. During interviews, the researcher involved in data collection took notes on participants' comments. A reflection note was taken immediately following each interview as well. The practices of bracketing and reflexivity were crucial to mitigating the potentially deleterious effects of unacknowledged preconceptions about the research and increased the rigour of the process. To ensure completeness and relevance, the PI verified all Chichewa-to-English transcripts. The findings were also triangulated, any gaps were discussed with the principal investigator, and the aggregated data was documented [19].

## Data analysis

**Quantitative data analysis.**    The extracted data were computed and cleaned in Microsoft Excel and imported for analysis into Stata v.14.0. The results from the descriptive statistics were expressed as percentages and frequencies for categorical data in tables and bar graphs. Metrics of central tendency like means, medians, inter-quartile ranges (IQR) and standard deviations for numerical data were calculated. The degree of correlation between the identified risk factors and neonatal mortality was assessed using univariable chi-square analysis. Multivariable logistic regression included risk factors that univariable chi-square analysis indicated had a significant connection after controlling for confounders. The adjusted odds ratios (AORs) and unadjusted odds ratios were calculated using logistic regression, with a significance level of 0.05 and a 95% confidence interval (CI). Imputation of severe missing variables

at rate of <5% or Complete Case Analysis (CCA) method was used to handle the missing data. Files were removed from the analysis.

**Qualitative data analysis.**   The qualitative data were manually analysed and used a thematic approach. A thematic analysis was conducted based on the data collected from key informants and in-depth interviews. A combination of Chichewa and English audios was verbatim translated into English, and interviews were immediately transcribed. The supervisors verified the transcriptions by editing them to ensure quality and correctness. The researcher initiated the familiarisation process while conducting the interviews. Later, they re-read the transcript multiple times to understand the depth of the content. Supervisors reviewed the code book and analysis guide during development. The data was analysed using inductive and deductive coding. The inductive codes were drawn from the collected data, while the theoretical codes were deduced from the interview guide and study objectives. Themes were identified from coded text segments that formed a similar, coherent, and familiar pattern, further organised to develop themes. Then the researchers reviewed and refined potential themes to match the rest of the data. Additional coding was done when the themes were incomplete or did not match the rest of the data or objectives. The thematic analysis allowed researchers to interpret and summarise themes for a report [21].

**Ethical approval and consent to participate.**   For the quantitative study, the study was approved by the University of Malawi College of Medicine Research Ethics Committee (COM-REC (12/22/3924) and the National Health Science and Research Ethics Committee (NHSRC (12/19/1180). For the qualitative study, participants provided written informed consent. The qualitative study was approved by COMREC (12/22/3924).

## Results

### Quantitative data

Three hundred eighty-three participants were screened for inclusion in the study, but only 237 were enrolled. Table 1 summarises the demographic characteristics of the mothers and neonates diagnosed with sepsis. The mother's median age was 24, with an interquartile range (IQR) of 9 (28–19). Most mothers were between 17 and 28, presenting 67% of the population. Of the 237 neonates diagnosed with sepsis enrolled in the study, 77% spent less than seven days in the hospital following admission, with a median stay in the neonatal care unit of 5 and (IQR) of 3 days.

In the univariable analysis, prematurity, mother's age, birth weight, gestational age weeks, type of birth (single vs multiple), and days spent in the hospital were highly correlated (P< 0.05) predictors of neonatal mortality refer to Table 2.

Prematurity was a statistically significant predictor of death (OR 4.71, 95% CI (1.713, 12.975), p value = 0.003), as was birth weight (OR 0.998, 95% CI (0.998, 0.999), p value = 0.001) refer to Fig 1.

Most deaths (11%) occur within seven days of admission refer to Fig 2.

On multivariable logistic regression analysis, after adjusting for prematurity effects, gestational age in weeks and hospital days were significant risk factors for neonatal sepsis-related mortality refer to Table 3.

### Qualitative data

Key informant interviews (KIIs) and in-depth interviews (IDIs) were conducted with ten healthcare professionals average age of 34 years and IQR of 6 years (28–22). Five HCWs had degrees, four had diplomas, and one had a certificate. The average interview duration was seventeen minutes. HCWs that were present during the data collection period interviewed

**Table 1. Characteristics of mothers and newborns admitted with sepsis to QECH.**

| Variables | Categories | Frequency (N = 237) | Percentage% |
|---|---|---|---|
| **Mother's age** | 1. 17–28 years | 160 | 67.0 |
| | 2. 29–39 years | 48 | 20.5 |
| | 3. ≥40 years | 22 | 9.6 |
| | 4. <17 years | 7 | 3.0 |
| **Mother's residence** | 1. Blantyre | 230 | 97.0 |
| | 2. Other | 7 | 3.0 |
| **ANC corticosteroids given** | 1. No | 134 | 56.5 |
| | 2. Unknown | 97 | 41.0 |
| | 3. Yes | 6 | 2.5 |
| **Maternal chronic conditions** | 1. None | 222 | 94.0 |
| | 2. Hypertension | 13 | 5.4 |
| | 3. Other | 2 | 0.8 |
| **VDRL status** | 1. Negative | 228 | 96.2 |
| | 2. Unknown/Positive | 9 | 3.8 |
| **HIV status** | 1. Negative | 193 | 81.4 |
| | 2. Positive | 32 | 13.5 |
| | 3. Unknown | 12 | 5.1 |
| **Type of birth** | 1. Singleton | 222 | 93.7 |
| | 2. Twins | 15 | 6.3 |
| **Sex** | 1. Male | 128 | 54.0 |
| | 2. Female | 100 | 42.0 |
| | 3. Unknown | 9 | 4.0 |
| **Birth location** | 1. Inborn | 142 | 60.0 |
| | 2. Other facilities | 75 | 31.6 |
| | 3. Other/Not recorded | 20 | 8.4 |
| **Reason for admission** | 1. Suspected sepsis/meningitis | 104 | 43.9 |
| | 2. Respiratory distress | 49 | 20.7 |
| | 3. Birth asphyxia | 33 | 14.0 |
| | 4. Small baby/preterm | 28 | 11.8 |
| | 5. Other | 23 | 9.6 |
| **Birth weight** | 1. >2500g | 145 | 61.0 |
| | 2. 1501-2500g | 54 | 23.0 |
| | 3. 1001-1500g | 21 | 9.0 |
| | 4. <1000g | 9 | 4.0 |
| | 5. Not recorded | 8 | 3.0 |
| **Gestation age** | 1. >37wks | 116 | 49.0 |
| | 2. 33-37wks | 94 | 40.0 |
| | 3. 28-32wks | 21 | 8.9 |
| | 4. <28wks | 6 | 2.5 |
| **Gestation methods** | 1. Fundal Height | 158 | 66.7 |
| | 2. Unknown | 43 | 18.1 |
| | 3. LMP | 21 | 8.9 |
| | 4. USS | 14 | 5.9 |
| | 5. Postnatal clinical gestational assessment | 1 | 0.4 |
| **Age** | 0–6 days | 237 | 100.0 |
| **Phototherapy** | 1. No | 193 | 81.4 |
| | 2. Yes | 44 | 18.6 |

(*Continued*)

**Table 1.** (Continued)

| Variables | Categories | Frequency (N = 237) | Percentage% |
|---|---|---|---|
| **Mode of delivery** | 1. Spontaneous vertex delivery (SVD) | 154 | 65.0 |
| | 2. Caesarean Section (CS) | 73 | 31.0 |
| | 3. Other/Unknown | 10 | 4.0 |
| **Blood culture results** | 1. Unknown result | 119 | 50.0 |
| | 2. Not done | 86 | 36.0 |
| | 3. Culture negative | 26 | 11.0 |
| | 4. Culture positive | 6 | 3.0 |
| **Vital signs checked** | 1. Yes | 233 | 98.0 |
| | 2. No | 4 | 2.0 |
| **Antibiotics given** | 1. Yes | 219 | 92.0 |
| | 2. No | 18 | 8.0 |
| **Cord chlorhexidine administered** | 1. Yes | 204 | 86.0 |
| | 2. No | 17 | 7.0 |
| | 3. Unknown | 16 | 7.0 |
| **Blood glucose test done** | 1. Yes | 216 | 91.0 |
| | 2. No | 21 | 9.0 |
| **Number of days spent in the hospital** | 1. <7 days | 183 | 77.0 |
| | 2. >7 days | 54 | 23.0 |
| **Discharge outcome** | 1. Alive | 216 | 91.0 |
| | 2. Dead | 21 | 9.0 |

**Table 2. Correlation between participant characteristics and mortality in newborns with sepsis.**

| Variables | Chi-square | P value | 95% confidence interval (CI) |
|---|---|---|---|
| Child age | 18.91 | 0.101 | (0.100, 1.226) |
| Mother's age | 6.70 | 0.025 | (0.918, 0.985) |
| Birthweight | 32.04 | 0.001 | (0.998, 0.999) |
| Gestational age | 30.23 | 0.001 | (0.653,0.831) |
| Type of birth | 4.45 | 0.020 | (1.260, 15.255) |
| Number of days spent in hospital | 17.81 | 0.001 | (0.553, 0.837) |
| Phototherapy | 0.00 | 0.953 | (0.308, 3.026) |
| Mode of delivery | 0.08 | 0.776 | (0.439, 3.017) |
| Birth location | 6.91 | 0.855 | (0.045, 0.897) |
| Blood glucose levels | 1.71 | 0.178 | (0.995, 1.029) |
| Sex | 0.13 | 0.716 | (0.481, 2.905) |
| Maternal chronic conditions | 0.19 | 0.992 | (0, 0) |
| HIV status | 6.62 | 0.261 | (0.154, 1.661) |
| Mothers residence | 1.46 | 0.173 | (0.474, 63.177) |
| VDRL status | 0.22 | 0.616 | (0.200, 15.195) |
| Blood culture results | 0.53 | 0.965 | (0.188, 4.952) |
| Cord chlorhexidine administered | 0.40 | 0.561 | (0.068, 4.284) |
| Reason for admission | 1.48 | 0.900 | (0.873, 8.543) |
| Use of ANC corticosteroids | 6.14 | 0.007 | (0.021, 0.544) |
| Prematurity | 18.91 | 0.003 | (1.713, 12.975) |

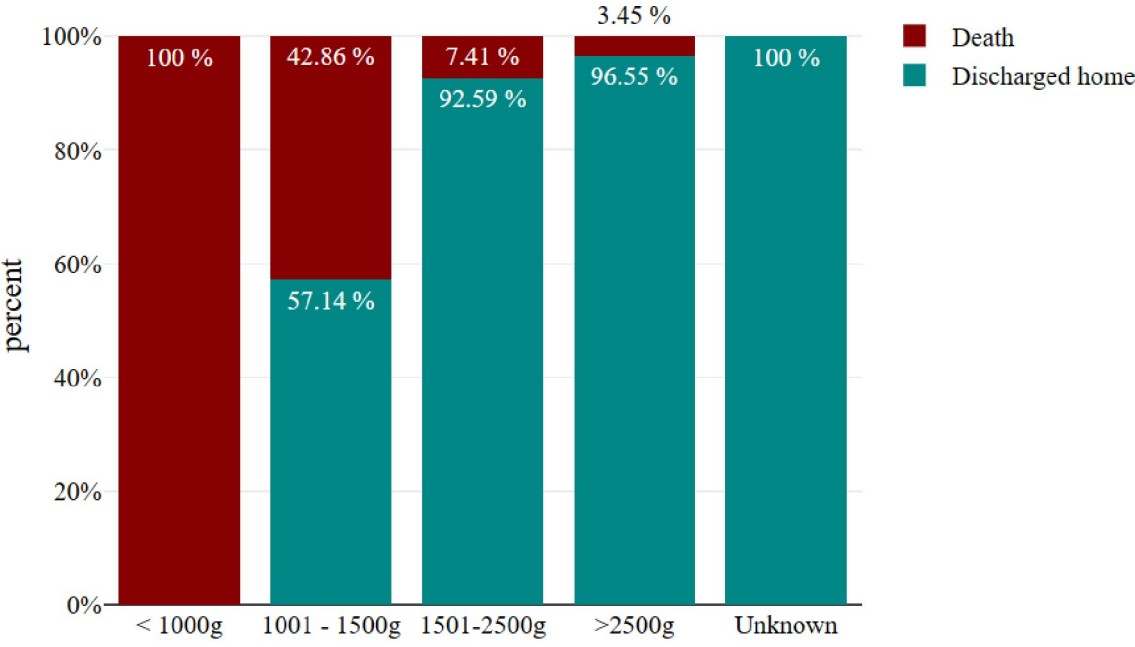

**Fig 1. Frequency of birth weight category.**

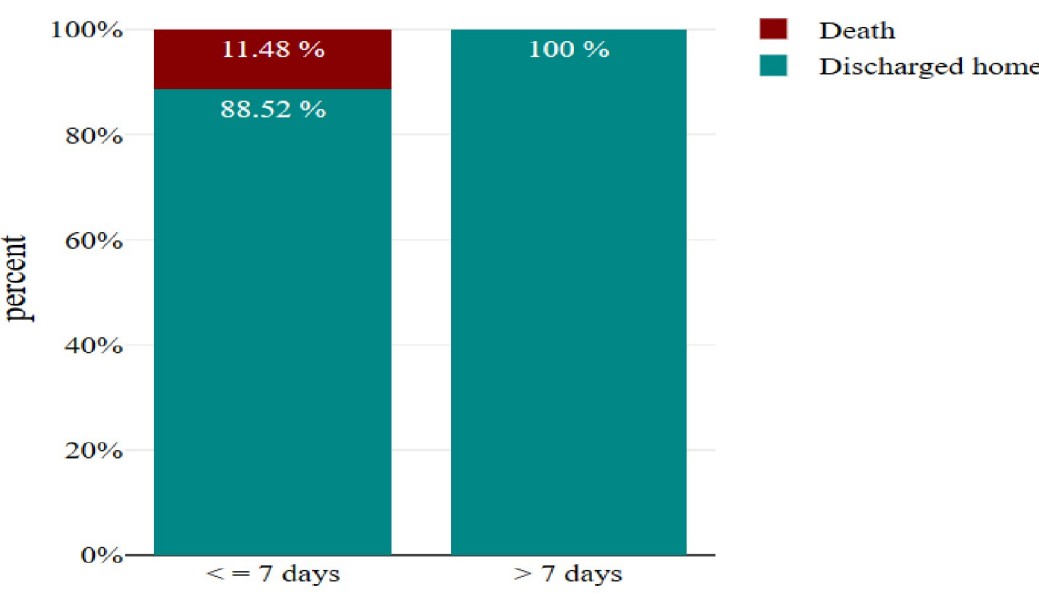

**Fig 2. Frequency of number of hospital stays duration.**

**Table 3. Multivariable logistic regression analysis model for death among hospitalised newborns with sepsis.**

| Predictor variables | N(%) | OR | (95% CI) | P-value |
|---|---|---|---|---|
| Prematurity | 182 (76.8%) | 0.348 | (0.041, 2.940) | 0.332 |
| Birthweight | 182 (76.8%) | 0.999 | (0.997, 1.000) | 0.085 |
| Gestational age | 182 (76.8%) | 0.760 | (0.584, 0.988) | 0.040 |
| Mothers age | 182 (76.8%) | 0.911 | (0.794, 1.046) | 0.187 |
| Type of birth (singleton vs. multiple) | 182 (76.8%) | 3.677 | (0.622, 21.730) | 0.151 |
| Number of days spent in hospital | 182 (76.8%) | 0.641 | (0.484, 0.851) | 0.002 |

included a matron, a ward in charge, nurses, and medical doctors. Work experience ranged from two months to 20 years.

## HCW perspectives of health systems factors contributing to neonatal sepsis-related deaths

*Maternal factors.* Health-related and behavioural factors were the two groups HCW cited as maternal factors causing neonatal sepsis fatalities. Maternal **health-related factors were prolonged labour, unnecessary vaginal examinations, a lack of monitoring, premature rupturing of membranes (PROM), maternal fevers, offensive liquor, and non-sterile equipment.** Participants discussed how premature rupturing of the membranes, spending much time in the labour ward before giving birth, and having unnecessary vaginal exams performed to check on the baby's progress made women more susceptible to infection, which then spread to newborns. Furthermore, mothers who gave birth at home or in transit put their newborns at risk of infection because they received assistance from unskilled birth attendants who used non-sterilized equipment.

> *"There may be maternal factors involved in the risk, such as high fevers before delivery, offensive liquor, assisted by unskilled birth attendants, or thick meconium-coloured liquor during delivery exposes the neonate to high chances of developing neonatal sepsis."*—**Nurse**

> *"Many babies are delivered at home and in transit, sometimes using contaminated equipment, while others are delivered in health centres and district hospitals. In these cases, it's possible that the mother experienced premature rupturing of the membranes (PROM), unnecessary vaginal examinations (VE), and spent too much time in the labour ward."*— **Medical doctor**

## Maternal behavioral factors included reporting late to the facilities, delivering at home, knowledge gaps, abscondment from the hospital, cultural and religious beliefs, and low socioeconomic level

Respondents reported that some parents refused to give their newborns injections because they believed sick newborns should not be given medication when ill. In addition, some mothers asked for discharge or absconded from the hospital before their newborns finished their treatments. They sometimes brought the newborn in late for treatment because they lacked knowledge of sepsis causes. While some guardians believed that applying dung caused the neonate's cord to heal faster, they did not know that doing so exposed the neonates to infections.

> *"Because mothers lack knowledge on the causes of sepsis, they sometimes bring their babies with fever to the hospital while covered with blankets instead of reducing beddings when a*

*neonate has a fever. Mothers tend to report late to the facilities when their neonates are sick. However, regardless of these factors, they are assisted when they bring the babies into the hospital for an early diagnosis."—**Nurse***

*"Of course, we have encountered guardians who wanted to take their neonates home for perhaps more traditional practices, so we run into that conflict where we sometimes have to counsel the guardians; maybe they won't allow; some abscond; and some we do say no; this is for the best of the baby; we have to keep the baby here despite the parents' insisting on going home. That's one barrier about cultural beliefs, too."—**Medical doctor***

*Neonatal factors*. The respondents mentioned **neonatal factors such as premature birth, babies born before arrivals (BBAs), meconium aspiration, overstaying in the hospital, and home deliveries** that also cause neonatal infections. The participants clarified that preterm neonates are susceptible to infections since their immune systems and lungs are not yet developed to handle such diseases. Sometimes, neonates aspirate thick meconium liquor during birth, predisposing them to infections. Since there is limited space for isolation, sick neonates are susceptible to hospital-acquired infections when they stay in the hospital long after admission.

*"Conditions like premature birth, BBAs, thick meconium aspiration, and possible long hospital stays come with exposure to other newborns with contagious infections."—**Medical doctor***

*"We usually receive neonates who had a delivery at home, babies born before arrival and in transit or the ambulance; sometimes they use unsterile equipment during delivery and delivered in a dirty place where the baby is exposed to infections."–**Nurse.***

*Health system factors*. The health system factors included **delays in neonatal care services cascade such as delayed referral, blood culture results and delayed treatment, poor attitude, lack of trainings, negligence and limited resources like equipment, space, drugs, tubings, workforce, reference materials, and follow-up visits.**

HCW narrated that blood samples were occasionally taken improperly or too late, potentially on day three, due to the increased workload in the ward. Due to the five- to seven-day wait for blood culture results, some respondents reported that it was difficult to diagnose sepsis with certainty, which delayed treatment. Some respondents believed that mothers arrived at hospitals later than they should have because they didn't know about the causes of neonatal sepsis. They also thought that neonates dying from sepsis increased due to the delayed referral of neonates with sepsis.

*"A delay in referring neonates with sepsis from other facilities is a contributing factor because they are brought in with a condition that has already worsened. Poor care from the mother may also occur if she is not properly instructed at the antenatal clinic or labour ward on how to care for the newborn. This is due to a knowledge deficit about sepsis causes."—**Medical doctor***

*"Blood culture sample is taken on admission and sent to the lab; getting the results takes 5 to 7 days. The neonates are commenced on first-line antibiotics, crystapen and gentamycin first and sometimes discharged without known blood culture results. Due to delayed results, doctors prescribe the right antibiotics late."—**Nurse***

Participants discussed how a lack of supplies, including spirits, detergents, sanitisers, nasal and gastric tubes, suction tubes, and medications, led to the spread of infections, which resulted in neonatal fatalities. In the pharmacy, there were instances when there were no

medications available. The other times, spirit and sanitiser were out of stock, and as a result, procedures were done without following aseptic techniques.

*"I will discuss resources. I forgot to mention one of the risk factors: we frequently reuse resources like nasal prongs and tubing. We do our best to decontaminate them fully, but this is not always possible, which can be one of the contributing factors. Similarly, antibiotics often experience inconsistent supplies. Sometimes, the pharmacy will not provide certain medications, but the department will purchase other antibiotics like Melopenam. There is a possibility that AAH health personnel will touch newborns without following standard protocols. This is if supplies like soap, spirits, and sanitisers aren't available."—**Nurse***

HCW stated that the staff shortage affects the unit's monitoring of neonates with sepsis. Other participants said that due to the high number of neonates admitted to the NCU with only a few staff members assigned to each shift and the lack of equipment like continuous monitors to monitor vital signs and alert HCW if there were any anomalies, neonates with apnoea were prioritised over neonates with sepsis.

*"Workload, um, shortage of staff. Sometimes we are very few, sometimes we are covered, but mostly we have few staff. Babies are missed not because we don't want to assist them but due to insufficient time and workload, so we prioritise those babies with apnoea rather than those with sepsis, and babies with sepsis are assisted later."—**Nurse***

HCW cited that the absence of adequate physical space for isolation resulting in congestion contributes to neonatal deaths. They further complained that there wasn't enough room for sepsis-stricken neonates to be isolated because there were too many unwell babies housed in one baby coat, which made infections easier to spread and increased neonatal fatality risk.

*"Mmm, I can say that sometimes we have cases where neonates need to be isolated, and we only have two isolation rooms, so it's difficult to isolate the babies. Sometimes guardians are also advised only to touch their babies, but they don't do that. Even health workers are advised to wash their hands after touching neonates, but occasionally they forget. One of the challenges is that we occasionally run out of medications required to treat newborns and equipment for wound dressing."—**Nurse***

Most participants mentioned that formal training, reference resources, and knowledge of neonatal sepsis were available. Healthcare professionals expressed varying opinions about formal training in newborn sepsis management. Some said they had taken such training, while others relied on their school experiences.

*"I cannot say that there is any formal training aside from that when I was coming, I was oriented on the protocol on management of neonatal sepsis and the other things like in-house training on how to collect blood culture, but not necessarily like there is training specifically for neonatal sepsis."—**Nurse***

Although the majority stated they had reference materials, other participants acknowledged that the unit lacked printed reference materials. Others explained that they have access to soft copies of standard operating procedures and protocols through their WhatsApp groups on their phones and books and posters on hand washing, chlorine dilution, and drug calculations that are pasted on the walls but not protocols for managing neonates with sepsis.

*"Yes, we have them. Some are pasted; others have soft copies. We shared it on the WhatsApp forum, where anyone can view it."*—**Nurse**

*"Mmm, yeah, books and charts on determining the proper dosage for medications like antibiotics are available in the ward." —***Nurse**

Participants discussed some of the infection control procedures followed in the NCU to stop infection spread and lower neonatal sepsis incidence. Participants agreed that scrubbing should be done at least once a month and highlighted that they would clean at least twice a month if resources were available. Damp dusting was done by moving the beds and disinfecting them with chlorine, soap, and clean water before neonates were placed on them. Before and after touching the neonates, while breastfeeding, and after changing diapers, guardians and HCWs in the unit were required to wash their hands. The guardians removed their shoes and caps when entering the unit. Hand sanitisers were installed and mounted on the walls for easy access by medical staff. Readmitted sick neonates and neonates with confirmed sepsis were isolated, and used equipment was decontaminated in chlorine and sterilised before use again.

*"Aah, infection prevention measures that we follow; we do monthly scrubbing at least once a month; we also do dump dusting; aah, we also make sure that when we have space, the neonates are not supposed to be mixed on the same bed, ideally because it is required one bed, one baby; and aah, if a baby is tiny and premature and needs KMC, we isolate them. If we have collected blood cultures and the results show that a particular organism is sensitive to some antibiotics, we also isolate those babies, and likewise, those babies that were discharged but are coming from home for readmission, we don't admit them here; we send them to the pediatric nursery." —***Nurse**

Lastly, HCW mentioned that follow-up visits and reviews were provided following discharge, especially for neonates with complications, and review dates were written in their health passport books. However, neonates who were discharged without complications were transferred to the postnatal ward and advised to come for a bench review in the unit whereby monitoring vital signs such as temperature takes place, the prescribed medications are given, and clinicians reviewed the neonates to see if any improvements. After completing their course of antibiotics, neonates were discharged and advised to report to the nearest health facility. Contact information was collected to ensure continuity of treatment for those discharged without receiving the blood culture results.

*"Okay, mostly when we discharge them, we don't allow them to come home straight; they are advised to report to postnatal care, where their mothers are after two days on antibiotics." "They are told to come for bench review, but after bench review, we discharge them home after health education on danger signs, and they are advised to go to the nearest facility for review."*–**Nurse**

## Discussion and recommendations

Our findings on the risk factors for neonatal mortality among hospitalised neonates with sepsis in the newborn care unit at QECH showed that gestation age and number of days spent in the hospital were the most important predictor factors in the multivariable logistic regression analysis model. The qualitative inquiry showed several themes that included health-related and

behavioural maternal factors, neonatal factors and health system factors, including prematurity, meconium aspiration, BBAs, home deliveries and prolonged stay in the facility; health system factors encompassing delays in neonatal care service cascade such as delayed referral, delayed blood culture results and delayed treatment, poor attitude, information, negligence and limited resources like equipment, space, drugs, tubings, workforce, reference materials, training and follow up visit. The theme that stands out the most is the delays in neonatal care service cascade such as delayed referral, delayed blood culture results and delayed treatment. This requires a multidisciplinary approach with interventions that could address these risk factors. Therefore, encouraging mothers in the community to use antenatal care services and early reporting to the facility might help identify the risk factors and possible interventions to reduce the risk factors of adverse birth outcomes, as well as reinforce neonatal sepsis education and promotion sessions targeting mothers and healthcare providers thereby reducing mortality [22].

The higher rate of deaths could be related to the reasons from the qualitative inquiry, such as prolonged labour, unnecessary VEs, lack of monitoring, PROM and use of unsterile equipment [18]. Similarly, to the studies done in the NCU in Mexico in 2015 [23], a public hospital in the Kaffa zone of southwest Ethiopia [2], Ruhengeri Referral Hospital, Rwanda [24] and North Ethiopia [25] that neonatal sepsis was linked to a history of maternal infections, genitourinary tract infection, the place of delivery, not received or received a single dose of tetanus toxoid injection, prolonged rupture of membranes, long labour, meconium-stained amniotic fluid, intrapartum fever, UTI/STI, and failure to breastfeed within one hour found to increase the odds of ascending microorganisms from the birth canal into the amniotic sac and fetal compromise as well as asphyxia, which frequently leads to sepsis in the neonatal period [6,27]. Therefore, encouraging mothers in the community to use antenatal care services and early reporting to the facility might help identify the risk factors and possible interventions to reduce the risk factors of adverse birth outcomes, as well as reinforce neonatal sepsis education and promotion sessions targeting mothers and healthcare providers thereby reducing mortality [22].

We found that neonates delivered at less than 37 weeks of gestational age were 0.76 times more likely to die within the first 28 days of life than those born at more than 37 weeks. This finding is consistent with previous studies in several African countries such as Ethiopia [26], Rwanda [27] and Kenya [23]. These results may suggest that immaturity of respiratory and cardiovascular organs and vulnerability to infection during the intrapartum and postpartum periods could be causing the risks in the newborns; therefore, preterm neonates' management should be emphasised in the facilities [8]. Improving the mother's nutritional status, using corticosteroids in preterm delivery, and avoiding high-risk behaviours such as smoking and alcohol intake during pregnancy is critical to improving birth weight. KMC is a valuable tool in managing these babies [26].

The length of stay in the NCU was significantly associated with mortality, and neonates who stayed less than seven days in the NCU were more likely to die than neonates who stayed more than seven days or more at the NCU. This has been previously reported in Ethiopia, where neonates who stayed for less than seven days in the NICU had a 3.9 times higher risk of neonatal mortality when compared to those who stayed for seven or more days. This can be explained because most neonatal deaths happen in the early neonatal periods (0–6 days of life) than in the late neonatal period of life (7–28 days) [22,23], and it may be related to the greater complexity of neonatal conditions at the NCU since this is a tertiary facility. The qualitative inquiry supports mothers bringing their neonates late to the facilities due to a knowledge deficit on the causes of newborn sepsis, cultural and religious beliefs and delayed referral of neonates with sepsis leading to increased mortality.

Furthermore, we found that deliveries is sometimes not aided by qualified and skilled personnel as a result care provided is compromised. Therefore, the care given needs to be improved by making sure that the deliveries are conducted by qualified skilled personnel. Similarly, a study conducted in Pakistan indicated that sub-standard care, inadequate training, low staff competence and a lack of resources, including equipment and medication, contribute to neonatal death [14].

For newborns, prolonged stay-NICU exposes them longer to the hospital environments, including the noise, bright light, and hospital-acquired infections, leading to a higher incidence of neonatal complications [27]. From the healthcare system perspective, prolonged length of stay -NICU could reduce the utilisation rate of beds and exacerbate the problem of inadequate healthcare resources. Therefore, with adequate medical support and monitoring, early discharge improves quality care while decreasing the overall cost of care. Beyond cost, early discharge programmes are also associated with high- satisfaction, preparedness, and positive quality indicators [28]. Strengthened linkages between households and HCWs are needed to support early discharge for stable newborns, with follow-up care provided by community HCWs at home and through outpatient visits to the facility.

The gaps in the provision of care arising from diagnostic, medication, disinfectants, equipment and staffing limitations, delays in blood culture results, reporting to facilities, lack of monitoring and lack of space for isolation even after NEST has invested in the facilities impede the provision of care and may contribute to mortality. The availability of resources such as formal training and reference materials enhances the management of neonates with sepsis. This finding is consistent with a research conducted in a rural district of eastern Uganda found that HCW also encountered difficulties in preventing the spread of infection to newborns within the facility due to understaffed and lack of essential or basic lifesaving equipment such as resuscitation kits for newborns [14]. Similarly, a qualitative research conducted in the QECH NCU, where caregivers found it extremely difficult to maintain proper hand hygiene due to a lack of essentials like linen and soap. An unreliable water supply for caregivers and medical staff made problems worse and increased the risk of sepsis in newborns [13].

Hospital management teams should ensure that infection prevention and control practices are followed. HCW should reinforce infection prevention practices during procedures, surgery, and childbirth. In addition, they should strengthen water and sanitation infrastructure, including adequate water and hand-washing facilities. Infrastructure should be renovated to create enough space for isolation and training for NCU employees. To improve blood culture diagnostic tests, the Ministry of Health must ensure equitable sepsis management and tests at all levels of healthcare (tertiary, district, and. health centre) to prevent and manage sepsis, provide timely referrals, and lobby with partners for resources.

Health facility managers must ensure that antiseptics, sanitisers, soap, and antibiotics are continuously available and available in sufficient quantities. In addition, instruments and supplies used in labour and childbirth must be sterilised and stored according to high standards. Community HCWs need to conduct campaigns to raise awareness about neonatal sepsis and encourage mothers to seek antenatal and postnatal care in the early stages of pregnancy. Detecting danger signs early will enable neonates to be treated appropriately in the event of a health problem.

## Study strengths and limitations

This mixed-methods study looked into potential risk factors for neonatal mortality in hospitalised newborns with sepsis. Both the quantitative and qualitative findings were relevant for the setting being based on the first hand experiences of the health care workers. Being a single-site

study made it cost-effective to implement. However, the study carried with it the limitations of a single-site study, which include overestimation and lack of generalization of the findings. Semi-structured interviews are often laden with subjectivity and consensus and external validity may be lacking. Additionally, due to missing data and the use of secondary data from routine data,the timing and duration of ART treatment for exposed neonates could not be accurately determined. The researcher employed study design bias control measures to mitigate the potential for bias in this study. Validation and triangulation of the data collected, helped to control for the study design bias. Randomization was controlled for sample selection bias. Other potential sources of are confirmation bias, interpretation bias during the researchers interpretation of the study data, and publication bias was evident in the researcher's reporting of both statistically and non-statistically significant results.

## Conclusion

Determinants of neonatal mortality in this study were primarily gestation age, number of days spent in the hospital, maternal behavioural and health related, neonatal and health system factors. Curbing the neonatal mortality among neonates with sepsis will require a multi-sectoral approach that includes interventions that address risk factors that could be clinical, health system and resources. These would have to be tackled at the same time to achieve reducing mortality.

## Supporting information

**S1 Appendix. Interview guide.**
(PDF)

## Acknowledgments

The authors would like to appreciate the QECH Paediatrics management, HCW in the NCU, for their numerous contributions that made this study a possibility.

## Author Contributions

**Conceptualization:** Lucky Mangwiro, Pui-Ying Iroh Tam, Kondwani Kawaza, Alinane Linda Nyondo Mipando.

**Data curation:** Lucky Mangwiro, Joseph Misyenje, Pui-Ying Iroh Tam, Kondwani Kawaza, Alinane Linda Nyondo Mipando.

**Formal analysis:** Lucky Mangwiro, Joseph Misyenje, Alinane Linda Nyondo Mipando.

**Investigation:** Lucky Mangwiro.

**Methodology:** Lucky Mangwiro.

**Project administration:** Lucky Mangwiro, Kondwani Kawaza.

**Supervision:** Pui-Ying Iroh Tam, Kondwani Kawaza, Alinane Linda Nyondo Mipando.

**Validation:** Lucky Mangwiro, Pui-Ying Iroh Tam, Kondwani Kawaza, Alinane Linda Nyondo Mipando.

**Visualization:** Lucky Mangwiro, Pui-Ying Iroh Tam, Kondwani Kawaza, Alinane Linda Nyondo Mipando.

**Writing – original draft:** Lucky Mangwiro.

**Writing – review & editing:** Pui-Ying Iroh Tam, Kondwani Kawaza, Alinane Linda Nyondo Mipando.

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
