## [Decision Letter · Decision Letter 0]

31 Jul 2024

PGPH-D-24-00266

Determinants of neonatal mortality among hospitalised neonates with sepsis at Queen Elizabeth Central Hospital, Blantyre, Malawi: A mixed-methods study

Dear Dr. Mangwiro,

Thank you for submitting your manuscript to PLOS Global Public Health. After careful consideration, we feel that it has merit but does not fully meet PLOS Global Public Health’s publication criteria as it currently stands. Therefore, we invite you to submit a revised version of the manuscript that addresses the points raised during the review process.

We look forward to receiving your revised manuscript.

Kind regards,

Lisa Miyako Noguchi

Academic Editor

Journal Requirements:

1. When completing the data availability statement of the submission form, you indicated that you will make your data available on acceptance. We strongly recommend all authors decide on a data sharing plan before acceptance, as the process can be lengthy and hold up publication timelines. Please note that, though access restrictions are acceptable now, your entire data will need to be made freely accessible if your manuscript is accepted for publication. This policy applies to all data except where public deposition would breach compliance with the protocol approved by your research ethics board. If you are unable to adhere to our open data policy, please kindly revise your statement to explain your reasoning and we will seek the editor's input on an exemption. Please be assured that, once you have provided your new statement, the assessment of your exemption will not hold up the peer review process.

Additional Editor Comments (if provided):

Please consider updating language that may be viewed as too subjective and/or judgmental (e.g., "filthy"). 

Reviewers' comments:

Reviewer's Responses to Questions

**Comments to the Author**

1. Does this manuscript meet PLOS Global Public Health’s publication criteria? Is the manuscript technically sound, and do the data support the conclusions? The manuscript must describe methodologically and ethically rigorous research with conclusions that are appropriately drawn based on the data presented.

Reviewer #1: Yes

Reviewer #2: Yes

2. Has the statistical analysis been performed appropriately and rigorously?

Reviewer #1: I don't know

Reviewer #2: Yes

3. Have the authors made all data underlying the findings in their manuscript fully available (please refer to the Data Availability Statement at the start of the manuscript PDF file)?

Reviewer #1: Yes

Reviewer #2: Yes

4. Is the manuscript presented in an intelligible fashion and written in standard English?

Reviewer #1: Yes

Reviewer #2: Yes

5. Review Comments to the Author

Reviewer #1: This is overall a very well thought out and executed study and a really useful first step in thinking about improving neonatal sepsis care at your hospital.

A couple of clarifications are needed in the article:

-In the methods, how was sepsis defined? Did you use a specific tool, local guidelines (if so what were the guidelines)

-In the methods, were interviews structured, semi-structured? How were the questions developed? It is fine if the questions were developed based on researcher/local expert consensus, but would be good to know (may be useful to share the questions in an appendix as well).

Methods: *Grammar check: Page 9 line 158 "The data was inductive and deductive coding."--it should probably say Data was analyzed using inductive and deductive coding

Discussion: Several themes were mentioned in the discussion as important factors based on the qualitative analysis--Was there one in particular that stood out or one that you would preferentially target?

-Line 434 in discussion, you talk about data from Pakistan-- is there data from sub-saharan Africa to suggest that there is sub-standard care being provided in NICUs (there are a few papers out there)? Did you notice this in your facility? I think throughout your discussion it is important to highlight that while your findings are consistent with other literature (which is valuable to note), it is also specific to your institution. Quality of care provision can vary greatly from one institution to another just within the same district, so it can be a bit of an overgeneralization to encompass all LMICs. So I would be cautious while making the point (while many global concerns may be the same, specific institutions have different outcomes).

Reviewer #2: Abstract well written, clearly shows the aim and objectives of the research. the methods and conclusion. the use of mixed methods was also good to explore both quantitative and qualitative figures. the writer maybe should have written how they got to 237 folders for the study. was there a power calculation that was done?

writer touched on global evidence and narrowed it down to Malawi and SSA. not sure i am clear as to how this study is related to the studies reviewed.

the tables were well presented. the results accurately captured the results

6. PLOS authors have the option to publish the peer review history of their article (what does this mean?). If published, this will include your full peer review and any attached files.

**Do you want your identity to be public for this peer review?** For information about this choice, including consent withdrawal, please see our Privacy Policy.

Reviewer #1: No

Reviewer #2: No

Additional (late arriving) comments from Reviewer #3 are also attached here and should be addressed by the authors.

---

## [Editor Report · Decision Letter 1]

22 Sep 2024

PGPH-D-24-00266R1

Determinants of neonatal mortality among hospitalised neonates with sepsis at Queen Elizabeth Central Hospital, Blantyre, Malawi: A mixed-methods study

Dear Dr. Mangwiro,

Thank you for submitting your revised manuscript to PLOS Global Public Health. After careful consideration, we feel that it has merit but does not fully meet PLOS Global Public Health’s publication criteria as it currently stands. Therefore, we invite you to submit a revised version of the manuscript that addresses one additional point raised during the review process.

We look forward to receiving your revised manuscript.

Kind regards,

Lisa Miyako Noguchi

Academic Editor

Journal Requirements:

**Editor Comment:**

**1. The discussion of limitations is still somewhat brief and would benefit from additional development. Please provide additional discussion on potential sources of bias, as well as the limitations of a single-site study. Consistent with guidance in the Standards for Reporting Qualitative Research, please also expand on researchers' characteristics that may influence the research (https://www.equator-network.org/reporting-guidelines/srqr/). **

---

## [Editor Report · Decision Letter 2]

26 Nov 2024

Determinants of neonatal mortality among hospitalised neonates with sepsis at Queen Elizabeth Central Hospital, Blantyre, Malawi: A mixed-methods study

PGPH-D-24-00266R2

Dear Miss Mangwiro,

We are pleased to inform you that your manuscript 'Determinants of neonatal mortality among hospitalised neonates with sepsis at Queen Elizabeth Central Hospital, Blantyre, Malawi: A mixed-methods study' has been provisionally accepted for publication in PLOS Global Public Health.

Please ask a senior member of your writing team to review the language in the limitations section. I am suggesting here some light revisions for clarity, but you should review these to make sure that they reflect your assessment of the study limitations:

// This mixed-methods study looked into potential risk factors for neonatal mortality in hospitalized newborns with sepsis. This methodology enabled the study to describe both quantitative and qualitative findings related to the experiences of healthcare workers. As a single-site study, it was cost-effective in terms of time and resources, but it may be subject to limitations as well, such as the lack of generalizability of the findings. Additionally, because this study used secondary data from a non-specific source, it was challenging to determine the start and duration of ART initiation for exposed neonates due to missing data. However, researchers employed study design bias control measures to mitigate the potential for bias in this study, including a mixed-methods study design to validate and triangulate the data collected, which may have controlled partially for study design bias. Random selection was also used to control for sample selection bias. More investigation would be required to determine additional infections that may have occurred because of the time spent in the hospital monitoring unclear blood culture results. //

Best regards,

Lisa Miyako Noguchi

Academic Editor
